# Data from the Paper Entitled "Application of a Bayesian Approach for Exploring the Impact of Syllable Frequency in Handwritten Picture Naming"

**CYRIL PERRET** (ID)

**CLARA SOLIER** (ID)

*Author affiliations can be found in the back matter of this article

Journal of
open psychology data

**DATA PAPER**

]u[ ubiquity press

## ABSTRACT

The data presented here comes from the Perret and Solier (2022) study. 30 participants handwrote labels for 150 black-and-white drawings. The experiment was carried out using the DmDx program. Response times and production errors were the two behavioral reported measures. DmDx scripts and data are available on the OSF platform (DOI: https://doi.org/10.17605/OSF.IO/GAZF3). These data should be useful for pre-testing to explore new hypotheses, as well as for methodological elements (e.g., sample size estimation, estimation of a priori distributions for Bayesian analyses).

**CORRESPONDING AUTHOR:**
**Cyril Perret**

Unversity of Poitiers, France

cyril.perret@univ-poitiers.fr

**KEYWORDS:**
Handwritten production; Response syllables; Error production

**TO CITE THIS ARTICLE:**
Perret, C., & Solier, C. (2024). Data from the Paper Entitled "Application of a Bayesian Approach for Exploring the Impact of Syllable Frequency in Handwritten Picture Naming". *Journal of Open Psychology Data,* 12: 5, pp. 1–7. DOI: https://doi.org/10.5334/jopd.110

# (1) BACKGROUND

The ability to use writing efficiently is one of the most important vectors of inclusion in modern societies. Writing skills are emancipation tools that give the individual power to transform and exercise their choices (Nussbaum, 2011; Sen, 1999). This fact has been reinforced in recent years by the development of digital communication tools, almost all of which require the possession of writing skills. Therefore, it seems important to explore the psychological mechanisms that enable a human being to use this modality of language production.

Writing involves first specifying the message the writer wishes to convey. This involves organizing ideas developed in the text during a conceptualization phase. Secondly, words need to be selected, organized according to rules of syntax, and spelled. This is the formulation phase. The third stage is dedicated to production. This involves choosing a medium (e.g., pen + paper, a computer or smartphone keyboard, etc.) from which the physical (paper) or virtual (screen) graphic trace will be created. Finally, during revision, the writer amends their texts by correcting errors, modifying certain parts, adding further information, and so on. These different stages in the text production, and their articulation, have been studied intensively (Olive, 2014).

Among works on writing, a part of them has focused on exploring mechanisms for accessing orthographic information and translating it into grapho-motor gestures (Perret & Olive, 2019). Numerous questions have been addressed, covering both the access to orthographic and grapho-motor representations stored in long-term memory (e.g., the syllable Perret & Solier, 2022 see below) and the way in which the stages of cognitive processing are articulated (e.g., Roux et al., 2013; Kandel & Perret, 2015). Studies are based on experimental tasks in which participants (adults as well as children) have to write isolated words from stimuli, i.e., images or words presented visually or aurally. Three types of behavioral measures are used to test hypotheses. Errors have been and still are a tool for exploring cognitive processing (e.g., Soum-Favaro, Solier, & Perret, in press). The other two measures of writing behavior are derived from time recordings: the time between the presentation of the stimulus and the initiation of response, known as response latency; and the time taken to produce the graphic trace, referred to as production duration (Perret & Olive, 2019).

The syllable is one of the linguistic units that have been the focus of research into the handwritten word production. In addition to findings from neuropsychology (e.g., Caramazza & Miceli, 1990; Badecker, 1996), there are two groups of experimental arguments in agreement with the hypothesis of syllable influence in the preparation of the orthographic verbal response.

On the one hand, if access to orthographic information involves retrieving syllables, the greater the number of units to be retrieved (e.g., 4 syllables vs. 2), the longer the response time should be. This influence of the number of syllables has been reported by Lambert and colleagues in triple copying tasks (2008, see also Sausset et al., 2012 and Lambert et al., 2015). On the other hand, the syllabic structure of a word influences the dynamics of grapho-motor tracing. Kandel et al. (2006) observed that the duration of crossing the inter-letter boundary (e.g., ac) is longer when it corresponds to a syllabic boundary (e.g., tra.ceur) than when it does not (e.g., trac.teur). This effect has been replicated many times (e.g., Alvarez et al., 2009; Hess et al., 2019; Kandel et al., 2011; Sausset et al., 2012).

A mental lexicon of orthographic syllables (i.e., a syllabary) has been proposed to account for these results (e.g., Kandel et al., 2011; Kandel, 2021). Since accessing each syllable within the syllabary is time-consuming, the preparation of a four-syllable word is expected to take longer than that of a two-syllable word. Furthermore, if this access process is carried out before a syllable is handwritten, the writer has to slow down the speed of the grapho-motor tracing due to the challenge of performing multiple activities simultaneously. The transition from one letter to another then takes longer when a syllabic boundary is involved. Finally, Sausset et al. (2013) demonstrated that the impact of the syllable occurred either during preparation or during grapho-motor execution, depending on the constraints inherent in the writing process.

The data presented in this article are derived from research aimed at testing the hypothesis of a mental syllabary in handwritten production (Perret & Solier, 2022). One major hypothesis regarding the functioning of the cognitive system suggests that the accessibility of information stored in long-term memory depends on the number of contacts and uses, i.e., its frequency of occurrence in the environment. Applying this reasoning to syllables, it should be possible to observe a frequency effect of this unit. The aim of the Perret & Solier (2022) study was to test this hypothesis in collecting the data of 30 participants who handwrote the labels of 150 images. The goal of the present work is to provide the behavioral responses, i.e., response times and errors collected in Perret and Solier (2022) study.

# (2) METHODS

## 2.1 STUDY DESIGN

Data were collected in a soundproofed experimental room. A within-participant design was employed: all participants handwrote the 150 pictures' labels. The items were divided into three lists. The order of drawings

was randomized within a list. Six orders of three lists were created. The six list orders were distributed among the 30 participants, i.e., 5 participants for a specific order. Each list begins with three trials to check that the participant has followed the instructions correctly.

An experimental trial proceeded as follows. A fixation cross ("+") appeared on the computer screen for 24 Time Frames (400 ms at 60 Hz) to fix the participant's attention. After a 6 Time Frames (100 ms at 60 Hz) black screen, the image appeared in the center of the screen and remained for 120 (2000 ms at 60 Hz) in the absence of a response. The first contact between the graphic tablet and the pen caused the item to disappear. The drawing was presented in reverse video mode (i.e., white lines on black screen) in a constant size of 9.5*9.5 cm. Finally, a black screen of 300 Time Frames (5000 ms at 60 Hz) separated two experimental trials, giving the participant time to produce the image label.

## 2.2 TIME OF DATA COLLECTION
The data were collected from November 25th, 2008 and April 8th, 2009.

## 2.3 LOCATION OF DATA COLLECTION
The data were collected in the University of Neuchâtel, Switzerland.

## 2.4 SAMPLING, SAMPLE AND DATA COLLECTION
The experiment was performed by 30 undergraduate students at the University of Neuchâtel. Participants were recruited in a speech therapy course on a voluntary basis. They received course credits for their participation. Among them, 10 were men. All participants were right-handed, native speakers of French and reported no handwritten language disorders. The mean age was 22.57 years (*SD* = 2.32).

## 2.5 MATERIALS/SURVEY INSTRUMENTS
Data were collected using the DmDx program (Forster & Forster, 2003; version 3.0.43) running on a laptop PC. Handwritten production was carried out on a sheet of paper with ninety-two 5-cm-long response lines distributed evenly across four columns. This sheet was placed on a graphic tablet (WACOM UltradPad A5, 200Hz). Participants handwrote using an inking contact pen (SP-401).

150 black-and-white drawings were selected from two French databases: Alario and Ferrand (1999) and Bonin et al. (2003). The ten pictorial and linguistic factors of images and their labels, used in the study by Perret & Solier (2022), are summarized in Table 1. Five factors—namely, Image Agreement, Image variability, Visual Complexity, Conceptual Familiarity, and Age of Acquisition—are typically measured using 5-point Likert-type scales with populations of young adults, often undergraduate students (Alario & Ferrand, 1999; Barry et al., 1997; Bonin et al., 2003; Snodgrass & Vanderwart, 1980). Image agreement (IA) is an estimate of the similarity between a subject's mental image generated by a word and the drawing used to represent the word (Barry et al., 1997) [1 = *very small (or null) degree of matching*; 5 = *very good match*]. Imagery variability (Ivar) estimates the extent to which a word evokes a greater or lesser number of mental images (Bonin et al., 2003) [1 = *few mental images*; 5 = *many mental images*]. Visual Complexity (VC) measures the degree of complexity of the drawing in terms of details (shape, surface details), features, their intricacy, etc. (Snodgrass & Vanderwart, 1980) [1 = *drawing very simple*; 5 = *drawing very*

| FACTORS | MEAN | Q1 | MEDIAN | Q3 | S.D. | MIN | MAX | SKEWNESS |
|---|---|---|---|---|---|---|---|---|
| NA(h) | .28 | 0.00 | 0.00 | 0.34 | 0.42 | 0.00 | 1.87 | 1.86 |
| IA | 3.67 | 3.20 | 3.72 | 4.25 | 0.74 | 1.23 | 4.90 | –0.63 |
| Ivar | 2.75 | 2.34 | 2.64 | 3.12 | 0.64 | 1.33 | 4.70 | 0.67 |
| AoA | 2.37 | 1.85 | 2.28 | 2.88 | 0.59 | 1.12 | 4.62 | 0.51 |
| LogFreq | 1.00 | 0.59 | 0.99 | 1.32 | 0.78 | 0.03 | 1.77 | 0.61 |
| NbLett | 5.85 | 4.00 | 6.00 | 7.75 | 1.98 | 2.00 | 10.00 | 0.09 |
| Fam | 3.06 | 2.13 | 3.07 | 4.02 | 1.09 | 1.03 | 4.97 | 0.02 |
| VC | 3.01 | 2.39 | 3.00 | 3.58 | 0.89 | 1.00 | 5.00 | 0.04 |
| LogMSyllF | 2.21 | 1.68 | 2.36 | 2.74 | 0.84 | –0.27 | 4.10 | –0.27 |
| LogFSyllF | 2.26 | 1.51 | 2.33 | 2.95 | 1.02 | –0.27 | 4.40 | –0.03 |

**Table 1** Descriptive statistics for the ten factors.

*Note. Q1* = 25th percentile; *Q3* = 75th percentile; *S.D.* = Standard Deviation; *Min* = Minimum; *Max* = Maximum; *NA(h)* = Name Agreement, H-statistic measures; *IA* = Image Agreement; *Ivar* = Image variability; *AoA* = Age of Acquisition; *LogFreq* = Natural logarithm of lexical frequency; *NbLett* = Number of letters; *Fam* = Conceptual Familiarity; *VC* = Visual Complexity; *LogMSyllF* = Natural logarithm of syllable Frequency; *LogFSyllF* = Natural logarithm of the first syllable frequency.

*complex*]. Conceptual familiarity (Fam) estimates how familiar a concept is within a language (Snodgrass & Vanderwart, 1980) [1 = *very unfamiliar concept*; 5 = *very familiar concept*]. Age of acquisition (AoA) refers to an adult's estimations of the age at which they acquired an image label (Bonin et al., 2003) [1 = *word acquired before 4 years*; 5 = *word acquired after 10 years*].

Name agreement (NA) refers to the extent of agreement among individuals in choosing a specific word to label or denote an image (Snodgrass & Vanderwart, 1980). NA is expressed using a measure of entropy (Lachmann, 1973), i.e., the h-statistic, indicating the greater or lesser number of alternative labels that can be used to name a drawing. Label length is measured by the number of letters (NbLett). The three frequency measures —namely, LogFreq, LogMSyllF and LogFSyllF— refer to the counting of the occurrences of an element within a corpus. The three raw frequency measurements were log-transformed (natural logarithm). Lexical frequency is the number of times a word appears in the FRANTEXT corpus (Lexique 2, New et al., 2004). Syllable frequencies are taken from the InfoSyll corpus (Chetail & Mathey, 2010). Finally, the lexical syllable frequency (LogMSyllF) of the word is obtained by averaging the frequency of each syllable in the word (Perret et al., 2014).

The behavioral measure selected by Perret and Solier (2022) is the latency of handwritten responses, i.e., the time between the presentation of the image and the first contact of the pen on the graphic tablet (RTs). The participant was instructed to handwrite (in lowercase) the name of the object represented by the picture presented on the computer screen, as quickly as possible while paying attention to the word spelling. The words had to be produced without their determiner. Only response times corresponding to the labels expected for an image with spelling in the canonical form were retained. If the participant did not recognize the object, s/he was asked to write DKO (don't know object) and ToT (Tip of the Tongue) if he recognized the object but could not find its name. This made it possible to remove the RTs corresponding to these situations. RTs corresponding to alternative labels (i.e., semantic errors) or spelling errors were also removed.

## 2.6 QUALITY CONTROL

To ensure proper understanding of the instructions, a set of three trials was carried out under the supervision of the experimenter (the first author). Moreover, the use of the graphic tablet could introduce a time factor, specifically the time needed to position oneself on the line dedicated to label production. To minimize this variability, participants were instructed to position themselves at the beginning of the line when the cross appeared for each item, with the pen slightly raised and ready to write. The experimenter ensured compliance

with this instruction during the training phase. Finally, participants were given a break after every 50 items to prevent excessive fatigue from impacting performance.

## 2.7 DATA ANONYMIZATION AND ETHICAL ISSUES

All participants provided written informed consent. The study complied with the 2008 Helsinki Declaration. The participant anonymization procedure involved assigning a unique code to each participant at the beginning of the study. This code was made up of the initial letters of the subject's first and last names, plus the age number. Finally, this code was converted into "Part" + number of participants in the data files.

## 2.8 EXISTING USE OF DATA

Data corresponding to response times (latencies) have been used in the Perret and Solier (2022) publication. The authors conducted comparisons using mixed-effects linear regression models, with parameters estimated through a Bayesian approach. The results indicated that syllable frequency had no influence on response times; in other words, the probability of observing the data was lower when the factor was included in the model. We refer the reader to the article for more information on statistical analysis and results. Error data are currently being used in a project on zero-inflation models.

# (3) DATASET DESCRIPTION AND ACCESS

## 3.1 REPOSITORY LOCATION

The data are available on OSF, Handwriting Syllable Frequency project (https://osf.io/gazf3/). A specific folder "Data Paper" contains all the documents. DOI of the project is https://doi.org/10.17605/OSF.IO/GAZF3.

## 3.2 OBJECT/FILE NAME

In the "Data Paper" folder, there are three folders. In the "Experiment" folder, three files correspond to the DmDx script: DmDx_Script1.rtf (https://osf.io/7pwf8); DmDx_Script2.rtf (https://osf.io/ezgm2); DmDx_Script3. rtf (https://osf.io/f2t9s). A README.txt file contains script metadata (https://osf.io/gbzms). In the "Data" folder, Material.csv (https://osf.io/us9de) lists all the items (in French and English) as well as their pictorial and lexical characteristics. Data.csv (https://osf.io/7skbd) corresponds to the behavioural measures. A README.txt file (https://osf.io/qntg4) presents the metadata of both Material.csv and Data.csv files. Finally, a "Pictures" folder contains the 150 black-and-white pictures.

## 3.3 DATA TYPE

DmDx scripts and the Material.csv file are primary data. The Data.csv file contains processed data.

### 3.4 FORMAT NAMES AND VERSIONS

DmDx script files are in Rich Format Text (.rtf) format. They can be read by Word (Office Suite) or used by DmDX. Data and Material files are in .csv format (Excel, Office Suite). The users can open those files on R-software (R Core Team, 2022) with the command read.csv2(file.choose()). Finally, the pictures are in bitmap format (.bmp).

### 3.5 LANGUAGE

The data is stored as American English. The experiment was run in French.

### 3.6 LICENSE

The open license is CC0 1.0 Universal.

### 3.7 LIMITS TO SHARING

N/A

### 3.8 PUBLICATION DATE

The dataset was published in the repository 18/01/2022.

### 3.9 FAIR DATA/CODEBOOK

As the data is stored on OSF, findability and accessibility are guaranteed. The project's DOI permits perennial identification and therefore findability. There are no identification or authorization procedures on OSF, thus guaranteeing accessibility. The presence of metadata for each file should enable good interoperability and reuse.

## (4) REUSE POTENTIAL

The data presented here correspond to response times and errors for a series of 150 black and white drawings handwriting by 30 participants. We believe that this database could be useful for researchers in psycholinguistics, linguistics and cognitive neurosciences interested in handwritten production, for several reasons.

This database is the result of an experiment whose experimental design is not based on a factorial design. Rather than creating two groups of items differing in syllabic frequency, with all other factors influencing behavioral measures equalized, a model-comparison approach was chosen, i.e., a regression model design. This choice of experimental design was justified by the fact that it is very difficult to control a large number of influencing factors while maintaining a reasonable sample size. Moreover, the factorial design, is generally highly questionable, as it is based on the absence of significance to null-hypothesis significance tests.[1] The use of regression model design offers the advantage of providing a database that can be reused to test new hypotheses. By operationalizing a new hypothesis with a new experimental factor, the same strategy of model comparison can be implemented. This can be an alternative to a time-consuming pre-test, for example. However, this possibility is limited by the language used. There is no a priori guarantee that the results obtained in French can be generalized to all languages.

This database also provides a tool for making certain a priori estimations. Works based on frequentist statistics are strongly advised to calculate participant sample sizes a priori (e.g., Cumming, 2014). From an ethical point of view, this allows researchers to mobilize only the number of participants needed to test a hypothesis. This calculation requires estimates of model parameters, in particular means and standard errors. This database offers the possibility of making these estimates. The same applies to the use of Bayesian statistics. One of the most delicate points in this type of analysis is to specify the characteristics of the a priori distributions of the model parameters (e.g., Kruschke, 2021). This database, supplemented using markov chains, can provide an estimate of the initialization parameters.

This database can also be used as a testing ground for statistical and probability works. It can be used to test processing algorithms on real data. It can also be used as a starting point for learning classification models for adult handwritten language disorders, for example. As data collection and processing are time-consuming activities, the database can be a starting point for this type of work.

Finally, the lack of influence of syllable frequency has only been observed once in the literature (Perret & Solier, 2022). It therefore seems important to attempt to replicate (or refute) this result. The availability of DmDx scripts is intended to facilitate these replications. In the same vein, this database could be included in a future meta-analysis on this type of question.

### NOTE

1   We refer readers interested in this methodological issue to the article by Sassenhagen & Alday (2016).

### ACKNOWLEDGEMENTS

The authors wish to thank Marina Laganaro, Patrick Bonin and Sonia Kandel for the scientific exchanges on this work and two anonymous reviewers for very helpful comments on a previous version of this paper.

### FUNDING INFORMATION

The data were collected as part of the project *"Integrating psycholinguistic, neurolinguistic and functional neuroimaging approaches in the study of normal and impaired language production, with focus on phonological encoding"* led by Marina Laganaro (Grant number: 118969).

## COMPETING INTERESTS

The authors have no competing interests to declare.

## AUTHOR CONTRIBUTIONS

CP: project supervision, study conceptualization, study design, data collection, data analysis, data curation, manuscript writing and editing.
CS: data analysis, data curation, manuscript writing and editing.

## AUTHOR AFFILIATIONS

**Cyril Perret** orcid.org/0000-0002-4552-9093
Unversity of Poitiers, France
**Clara Solier** orcid.org/0000-0001-8602-1205
Basque Center on Cognition, Brain and Language, Spain

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

**Soum-Favaro, C., Solier, C.,** & **Perret, C.** (in press). The analysis of errors in written word and sentence production: The value of a classification for French. *Written Language and Literacy*.

## PEER REVIEW COMMENTS

*Journal of Open Psychology Data* has blind peer review, which is unblinded upon article acceptance. The editorial history of this article can be downloaded here:

- **PR File 1.** Peer Review History. DOI: https://doi.org/10.5334/jopd.110.pr1

**TO CITE THIS ARTICLE:**
Perret, C., & Solier, C. (2024). Data from the Paper Entitled "Application of a Bayesian Approach for Exploring the Impact of Syllable Frequency in Handwritten Picture Naming". *Journal of Open Psychology Data,* 12: 5, pp. 1–7. DOI: https://doi.org/10.5334/jopd.110

**Submitted:** 20 February 2024　　**Accepted:** 17 April 2024　　**Published:** 06 May 2024

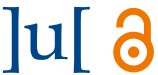