## [Peer Review History. · Journal of Open Psychology Data]

Dear Editor, Dear Dr Evans,

First of all, we'd like to thank you for giving us the opportunity to review the manuscript *Data from the paper entitled "Application of a Bayesian approach for exploring the impact of syllable frequency in handwritten picture naming"*. We have considered all the comments made by the two reviewers. We detail each point below. We have also included the answers to the questions and made the modifications in the manuscript directly. They are highlighted in blue in the body of the text.

We hope that these revisions will enable you to accept our manuscript for publication.

Cyril Perret & Clara Solier

Reviewer 1:

This paper presents data on handwritten picture naming, including response time and error measures. The data was assessed from 2008 to 2009 in Switzerland. The authors suggest that it can be used for linguistic and methodological research.

In accordance with the review guidelines, the data are deposited in a suitable repository (OSF) under an open license that permits unrestricted access (CC0). The data are findable, accessible, interoperable, and reusable. The data are comprehensible due to sufficient labels and a README file with metadata provided in the same repository. All files (data, scripts, metadata) are available in English and can be accessed by common computer programs (e.g., Word, Excel, R).

The paper is well written and contains detailed information about the published data. However, I have few suggestions for further improvement (see also uploaded Word file):

Q1. Please provide more information on the ten pictorial features (e.g., definition, how was the feature assessed) in section 2.5.

A1. We have included a description for each factor, as well as details of the databases from which the measurements originated and how they were collected.

Q2. If available, you could provide more information on the sampling strategy (how were the participants recruited?) in section 2.4.

A2. We had no recruitment strategy. We asked students who had just completed a logotherapy course if they had a little time to help us with psycholinguistic research. We added these elements to the manuscript.

Q3. I feel that the current paper contains too much information on the paper that reports the analyses and results based on the proposed data (Perret & Solier, 2022). The current paper is mainly supposed to provide readers with information on the data and not to summarize the papers that base their analyses on the data at hand. This observation concerns most sections, e.g. the abstract, the background, sections 2.4 and 2.5, and the reuse potential section. The respective passages are highlighted in green in the uploaded Word file. Consider removing the highlighted passages, especially removing information on the specific hypotheses, the analyses (e.g. which statistical models and other techniques were used) and results of Perret & Solier, 2022. Instead, you could shortly summarize the hypotheses, analyses and findings in section 2.8 and refer to the original paper for more information.

A3. We followed the reviewer's recommendations. References to statistical analyses and the results of the Perret & Solier (2022) study (highlighted in green for the reviewer) have been

removed. We have added some information on these aspects in section 2.8. A cross-reference has been made to the article.

Q4. I have further some minor comments regarding the language and formalities (the respective passages are highlighted in yellow in the uploaded Word file):

(1) Please consider checking the tense, punctuation and spacing: sometimes, there were double punctuations or spacings or no spacing where there should be one, or sentences were in present tense when past tense was needed.

(2) In section 2.4, consider rewriting the first sentence to “Thirty speech therapy students at the University of Neuchâtel (Bachelors 1, 2 & 3) participated in the study.” to avoid doubling the term participants/participated. In the same sentence, I am not sure what you mean with Bachelors 1 & 2 & 3. Please elaborate on this. Furthermore, the second sentence of section 2.4 is redundant with a part of the fifth sentence (native speakers of French). Consider removing the second sentence. Lastly, regarding to APA, statistical terms (e.g., SD) should be written in italics and with a space before and after “=”.

(3) In section 3.4, consider rewriting the last sentence to “... readable with Excel (Office Suite) and the statistical software R (command: ...)”. Additionally, insert a “.” at the end of the paragraph. Also consider including a reference for R.

A4. Thank you for pointing out these various typos, errors, etc. We have corrected them and proofread the entire manuscript.

Reviewer 2:

The contents of the paper and the deposited data fulfilled most of the requirements, except for the following:

Q1. In the abstract <https://osf.io/gazf3/>; DOI 10.17605/OSF.IO/GAZF3: the first one is an OSF link, what is the following part (it's a not a searchable DOI)?

A1. <https://doi.org/10.17605/OSF.IO/GAZF3> is the project DIO. It can be consulted directly. The link in the summary did not work because the beginning of the link was missing. We have corrected this error.

Q2. For easier access, please provide an OSF link directly to the data that's reported here instead of the entire repo of the project.

. [This is connected to the above point] "DmDx scripts .. are available on the OSF platform": I couldn't find any 'DmDx scripts'; either point directly to them or have transparent filenames (e.g. DmDx_script_1, DmDx_script_2 etc.)

. [This is also connected to the same point] "The scripts for the three lists are available in .rtf format in the "Data Paper" folder": The scripts are actually inside another folder inside the 'Data Paper' folder.

A2. We chose to use the warehouse link when referring to all the elements made available. We then added direct links in the manuscript for each specific element (data, material, etc.). Thank you for making us aware that access to information was not very clear.

Q3. "One hundred and fifty black-and-white drawings were selected from two French databases": Shouldn't the selected drawings also be included in the OSF dataset?

A3. We've added a folder “Pictures” with all 150 images.

Q4. "a file (material.csv)", "second file (data.csv)", "the "material" file", "The Data file":
Please use actual and correct filenames from the OSF repo (Data.csv, Material.csv).

A4. We have homogenized the names in line with those used in the OSF warehouse.

Q5. Some minor corrections, typos etc. (in general, please proofread the text). There are many typos, here are a few of those (maybe try using Google docs which easily catches many of these typos and also some grammatical mistakes):

- . "labels for one hundred and fifty" -> labels for 150 (150 at multiple places); thirty participants -> 30 participants
- . "Keywords ... Error production;" -> "Keywords ... Error production" (trailing ';')
- . itseems -> it seems
- . Baddecker, 1996 -> Badecker, 1996
- . Found at least two instances of hanging (unmatched) double quotes on pp. 3-4
- . Typos: "participation.Among them, 10 were men.. "; "to label production.. To minimize"; "the study.. This code"
- . "running on laptop PC" -> running on a laptop PC; "the pen causes item to disappear" -> the pen causes the item to disappear
- . "If the participant did not recognize the object, he was" -> If the participant did not recognize the object, s/he was

A5. Thank you for pointing out these various typos, errors, etc. We have corrected them and proofread the entire manuscript.